# Lipopolyplex-Mediated Co-Delivery of Doxorubicin and FAK siRNA to Enhance Therapeutic Efficiency of Treating Colorectal Cancer

**DOI:** 10.3390/pharmaceutics15020596

**Published:** 2023-02-10

**Authors:** Tilahun Ayane Debele, Chi-Kang Chen, Lu-Yi Yu, Chun-Liang Lo

**Affiliations:** 1Department of Biomedical Engineering, National Yang Ming Chiao Tung University, Taipei 112, Taiwan; 2Department of Chemical & Environmental Engineering, College of Engineering and Applied Science (CEAS), University of Cincinnati, Cincinnati, OH 452, USA; 3Medical Device Innovation and Translation Center, National Yang Ming Chiao Tung University, Taipei 112, Taiwan

**Keywords:** polyplex, lipopolyplex, gene therapy, chemotherapy, cancer therapy, pyridoxal

## Abstract

Tumor metastasis is a major concern in cancer therapy. In this context, focal adhesion kinase (FAK) gene overexpression, which mediates cancer cell migration and invasion, has been reported in several human tumors and is considered a potential therapeutic target. However, gene-based treatment has certain limitations, including a lack of stability and low transfection ability. In this study, a biocompatible lipopolyplex was synthesized to overcome the aforementioned limitations. First, polyplexes were prepared using poly(2-Hydroxypropyl methacrylamide-*co*-methylacrylate-*hydrazone*-pyridoxal) (P(HPMA-*co*-MA-*hyd*-VB6)) copolymers, which bore positive charges at low pH value owing to protonation of pyridoxal groups and facilitated electrostatic interactions with negatively charged FAK siRNA. These polyplexes were then encapsulated into methoxy polyethylene glycol (mPEG)-modified liposomes to form lipopolyplexes. Doxorubicin (DOX) was also loaded into lipopolyplexes for combination therapy with siRNA. Experimental results revealed that lipopolyplexes successfully released DOX at low pH to kill cancer cells and induced siRNA out of endosomes to inhibit the translation of FAK proteins. Furthermore, the efficient accumulation of lipopolyplexes in the tumors led to excellent cancer therapeutic efficacy. Overall, the synthesized lipopolyplex is a suitable nanocarrier for the co-delivery of chemotherapeutic agents and genes to treat cancers.

## 1. Introduction

Among potential applications for cancer therapy, combinational therapy that combines chemotherapy and gene therapy has attracted considerable interest in recent times [1,2,3]. Several gene therapy studies have explored the use of siRNAs to silence the expression of oncogenes and to target specific mRNAs that promote the proliferation of cancer cells through their complementary pairing and cleavage [4,5,6]. Doxorubicin (DOX) is one of the most commonly used chemotherapy drugs for treating several cancers. Recently, new strategies of cancer co-therapy combine DOX with a gene that can modulate the tumor immune responses or drug resistance to improve the therapeutic efficiency of DOX [1,7]. More importantly, drug resistance is highly associated with CXC chemokine receptor 4 (CXCR4) and focal adhesion kinase (FAK) to up-regulate resistance protein expression [8]. FAK is a 125-kDa nonreceptor tyrosine kinase found at adhesion sites between cells and extracellular environments [9]. FAK acts as a primary regulator of focal adhesion signaling to regulate cell survival, proliferation, invasion, and migration [10]. Overexpression and/or increasing activity of FAK has been detected in primary and metastatic cancers and is related to poor clinical outcomes, such as in ovarian, breast, pancreatic, lung, melanoma, prostate, colorectal, glioblastoma, and esophageal cancers [11,12,13]. Hence, selective inhibition of FAK using FAK siRNA could be a potential target for cancer therapeutics. However, naked siRNA injected into the body has a short half-life, limiting its usefulness.

In addition, chemotherapy is the most widely used method to treat cancers, even though its success is limited owing to the development of drug resistance (specifically, reduced cellular uptake owing to modifications of membrane lipids, enhanced efflux pumps that decrease the cellular concentration of the drug, altered drug targets, repair of the damaged DNA, and inhibition of apoptosis) after repeated administration [14,15,16]. DOX, a well-known anthracycline antibiotic known for its anti-cancer activity, has been widely used to treat several types of cancer. DOX intercalates with DNA base pairs (i.e., it disrupts topoisomerase-II-mediated DNA repair) and targets multiple molecular targets to produce a range of cytotoxic effects (e.g., generation of free radicals that damage cellular membranes, DNA, and proteins) [17,18]. However, this popular anti-cancer drug has certain limitations, such as a lack of selectivity, low bioavailability, and adverse side effects on normal cells/tissues [19,20]. Furthermore, the FAK signal pathway not only regulates the epithelial–mesenchymal transition but is also highly associated with the drug resistance-related genes ABCB1 and ABCC1, overexpression of DOX-resistance, and paclitaxel resistance [21,22,23].

Hence, it is vital to develop novel nanocarriers for optimizing the therapeutic effect of drugs on cancer cells while minimizing their side effects on normal cells. Several studies have shown that combining anti-cancer drugs with siRNAs could be the best tactic to reduce the dose-related side effects of anti-cancer drugs [24,25,26]. However, the primary challenge in the co-delivery of anti-cancer drugs and siRNAs is to select the best carriers, given the differences in the physical properties of anti-cancer drugs and siRNA. Therefore, in order to overcome the aforementioned shortcomings, several researchers have focused on the development of non-viral gene delivery vectors such as cationic liposomes, cationic polymers, peptides, and dendrimers [27,28,29]. Of these, lipopolyplexes, composed of lipid, polycation, and nucleic acid, are a well-known nanocarrier showing superior colloidal stability, less cytotoxicity, and high gene transfection efficiency as it combines the advantages of polyplexes and lipoplexes. Several reports have shown that lipopolyplexes consisting of polyethyleneimine and cationic liposomes could enhance in vitro transfection efficiencies and improve serum stability, even though their high toxicity remains a major drawback [30,31,32]. As a result, lipopolyplexes containing neutral, anionic, or polyethylene glycol (PEG)-modified phospholipids are considered to be highly promising candidates.

Although many nanocarriers have been developed to simultaneously deliver DOX and siRNA, the retention time of siRNA in intracellular endosomes is too long to reduce its therapeutic efficacy. Therefore, in this study, a novel lipopolyplex composed of a core of charge-converted copolymer-condensed FAK siRNA coated with a lipid bilayer and mPEG outer shell was investigated (Figure 1). Pyridoxine (Vitamin B6, VB6) is one of the most versatile enzyme cofactors, and its active form, pyridoxal 5′-phosphate, is a bioactive coenzyme involved in approximately 4% of all classified enzymatic reactions [33,34]. VB6 plays a crucial role in multiple aspects of cellular activity and is safe for the human body. Pyridoxal is a VB6 derivative; the pKa values for the hydroxyl group and nitrogen in the pyridine are 4.23 and 8.7, respectively [35]. Hence, pyridoxal exhibits a neutral zwitterionic structure at pH 7. In this study, the synthesized charge-converted copolymer, poly(2-Hydroxypropyl methacrylamide-*co*-methylacrylate-*hydrazone*-pyridoxal) (P(HPMA-*co*-MA-*hyd*-VB6), bore a positive charge at low pH value owing to protonation of pyridine of pyridoxal and electrostatically interacted with negatively charged FAK siRNA to form polyplexes. Furthermore, the synthesized polyplexes and DOX were encapsulated into liposomes for the co-delivery of DOX and FAK siRNA to treat colon cancer cells. Given that the hydroxyl group in the pyridine could absorb protons at low pH, FAK siRNA, and DOX could escape from lipid bilayer-constructed vehicles (liposomes and endosomes) via a proton sponge effect and then be transported into the cytoplasm (Figure 1). Here, the influences of N/P ratio, pH, and the molecular weight/pyridoxal contents of polymers on polyplex formation were investigated to obtain an optimal polyplex. Furthermore, DOX-loaded lipopolyplexes were formed and tested for their functions in in vitro as well as in vivo models to evaluate the feasibility of using such lipopolyplexes in cancer therapy.

## 2. Materials and Methods

### 2.1. Materials 

1,2-Dipalmitoyl-sn-glycero-3-phosphocholine (DPPC) was purchased from Avanti (Birmingham, AL, USA). Cholesterol, Trifluoroacetic acid (TFA), and Chloroform-d (CDCl_3_), and Dimethyl sulfoxide-d_6_(DMSO-d_6_) were purchased from ACROS (Geel, Belgium). *tert*-butyl carbazate, Boc-hydrazide, methoxy poly (ethyl glycol) 5000 (mPEG-5000), Doxorubicin hydrochloride (DOX-HCl), pyridoxal hydrochloride (VB6-HCl), Thiazolyl blue tetrazolium bromide, N,N’-Dicyclohexylcarbodiimide (DCC), and 3-mercaptopropionic acid (MPA) were purchased from Sigma-Aldrich (St. Louis, MO, USA). N-(2-Hydroxypropyl) methacrylamide (HPMA) was purchased from Polysciences (Warrington, PA, USA). 4-Dimethylaminopyridine (DMAP), Succinic anhydride, and Methylacrylic chloride (MACl) were purchased from Alfa (Indianapolis, IN, USA). 2,2′-Azobis(2-methylpropionitrile) (AIBN) was purchased from UniRegion Bio-Tech (Taoyuan, Taiwan). Dichloromethane (DCM), Diethyl ether, Ethyl acetate, and Dimethyl sulfoxide (DMSO) were purchased from ECHO (Miaoli, Taiwan). FAK siRNA [sense:5′-AACCACCUGGGCCAGUAUUAU-3′; antisense: 5′-AUAAUACUGGCCCAGGUGGUU-3′], FAM-FAK siRNA, [sense:5′-AACCACCUGGGCCAGUAUUAU-3′; antisense: 5′-AUAAUACUGGCCCAGGUGGUU-3′] were purchased from MDBio, Inc. (Taipei, Taiwan). Other reagents were of analytical grade and used without further purification. Distilled water used in all the experiments was purified using an AquaMax-Ultra water purification system (Younglin Co., Anyang, Korea).

### 2.2. Synthesis of Methoxy Poly (Ethylene Glycol)-Cholesterol (mPEG-Cholesterol)

Methoxy poly (ethylene glycol)-cholesterol was synthesized via a two-step reaction. First, mPEG5000 (1 mmol) was reacted with succinic anhydride (3 mmol) in the presence of a small amount of DMAP at room temperature under nitrogen for 24 h. After the reaction, the mPEG5000-COOH product was precipitated from diethyl ether three times. The structure of mPEG5000-COOH was confirmed using ^1^H-NMR (Bruker NMR 400 MHz, Billerica, MA, USA) and Fourier transform infrared spectroscopy (FT-IR, SHIMADZU IRAffinity-1, Kyoto, Japan). Second, mPEG5000-COOH (1 mmol) was dissolved in DCM and mixed with cholesterol (1.5 mmol) and DMAP/DCC (0.3 mmol) in a two-necked flask. The reaction was continued under nitrogen and stirred well for 24 h. After the reaction, byproducts and residue cholesterol were removed using gravity filtration and precipitation in diethyl ether three times. The product was completely dried and identified by ^1^H-NMR and FT-IR.

### 2.3. Synthesis of Pyridoxal Containing Polymers (P(HPMA-co-MA-hyd-VB6))

Before polymer preparation, MABH monomer was synthesized using methacrylic chloride (MACl) and *tert*-butyl carbazate (Boc-hydrazide). The detailed synthesis protocols are described in the Appendix A. Pyridoxal (a vitamin B6 derivative) containing the polymer P(HPMA-*co*-MA-*hyd*-VB6) was then synthesized through a two-step reaction. First, P(HPMA-*co*-MABH) polymer was synthesized using free radical polymerization. Various proportions of AIBN, HPMA, MABH, and MPA were dissolved in DMF or MeOH under nitrogen. Furthermore, the mixture was placed in a 70 °C oil bath, and free-radical polymerization was continued under stirring for 24 h. After the reaction, the solution was precipitated in diethyl ether three times to obtain P(HPMA-*co*-MABH) polymers. The above-synthesized polymer was dissolved in TFA/DCM (*v*/*v* = 1:1) and reacted at room temperature for 3–4 h to remove the protective groups. It was then dried using a rotary evaporator to obtain P(HPMA-*co*-MA-NH_2_) polymers. Second, the P(HPMA-*co*-MA-NH_2_) polymers were dissolved in DMSO (2 mL) and mixed with a 1.5-fold molar of pyridoxal hydrochloride (2 mL H_2_O). The reaction continued at 50–60 °C for 48 h under stirring. After the reaction, free pyridoxal hydrochloride was removed using dialysis (MWCO 1000Da) against DMSO/DIW (*v*/*v* = 1:1) for 3 days, followed by lyophilization to obtain VB6 conjugated polymers, P(HPMA-*co*-MA-VB6) products. P(HPMA-*co*-MA-*hyd*-VB6) and its precursors were characterized using 1H-NMR and FT-IR. The FT-IR sample was prepared by mixing the polymer product with KBr at a ratio of 1:50 (*w*/*w*). In addition, GPC analysis was performed using a Shodex SB 804-HQ column (8 mm ID × 300 mm L and 9 μm, particle size, Tokyo, Japan) to confirm P(HPMA-*co*-MA-*hyd*-VB6) formation. A standard PMMA was used to calculate the molecular weight of P(HPMA-*co*-MA-*hyd*-VB6). The analysis was conducted using 50 mM LiBr in DMF as a mobile phase at a flow rate of 1 mL/min.

### 2.4. Preparation and Analysis of Polyplexes

First, P(HPMA-*co*-MA- *hyd*-VB6) polymers were completely dissolved in DMSO/pH 7.4 PBS solution (*v*/*v* = 1:1) or DMSO/pH 4.0 300 mM citrate buffer (*v*/*v* = 1:1). The polymer solution was diluted to the required volume and mixed with 10 μL of 20 μM siRNA to become polymer-siRNA mixture solutions (N/P ratios from 200:1 to 1:1). After shaking at room temperature for 20 min, the solution was mixed with 0.25% bromophenol blue and 30% glycerol aqueous solution, loaded onto 2% agarose gels, and run at 80 V for 30 min. After that, the gel was stained with ethidium bromide and photographed using a UV photo-adhesive system.

### 2.5. Preparation of Lipopolyplexes (LP) and DOX-Loaded Lipopolyplexes (DLP)

To prepare LP, phospholipid (DPPC) and cholesterol (molar ratio 5:4) were homogeneously dissolved in MeOH/DCM (*v*/*v* = 1:2). The organic solvent was removed through rotary evaporation to prepare a thin film layer. Furthermore, the thin film was hydrated using 1.5 mL of polyplex-containing citrate buffer (pH 4.0). A 4-fold volume of diethyl ether (aqueous: diethyl ether = 1:4 (*v/v*)) was then added, followed by emulsification using a sonication probe for 10 min. After removing the diethyl ether using rotary evaporation, mPEG-cholesterol in citrate buffer (pH 4.0) was added and incubated at 60 °C for 1 h. The solution was filtered by an extruder having a 200-nm polycarbonate membrane 11 times and used the Sephadex G-50 column (Sigma-Aldrich, St. Louis, MO, USA) with a pH 7.4 phosphate buffer solution to remove free polymers and prepare lipopolyplexes (LPs). To prepare DOX-loaded LP (DLP), the pH value of the LP solution was adjusted to 7.4 after column chromatography, followed by shaking with DOX solution (1 mg/mL) for 1.5 h. The DOX was then encapsulated in the core of the DLP. The unloaded DOX was removed using column chromatography (Sephadex G-50 column). The particle size, zeta potential, and polydispersity index for LP and DLP were determined by dynamic light scattering (DLS) using a Malvern Zetasizer Nano S apparatus (Worcestershire, UK) equipped with a 4.0 mW laser operating at λ = 633 nm and a scattering angle of 90°. All measurements were performed at 25 °C, and the data were obtained from the average of three measurements. The morphology of LP and DLP stained by 2% of PTA was analyzed using transmission electron microscopy (TEM, using a JEOL JEM-2000EX instrument at a voltage of 200 kV, Tokyo, Japan). To investigate the drug content and encapsulation efficiency of the DLP, the prepared DLP was dissolved in DMSO, and then the absorbance of DOX was measured using a UV-Vis spectrophotometer (UV-Vis, Beckman DU-800, Brea, CA, USA) at 480 nm. The percentages of drug content and encapsulation efficiency were calculated using the following formula:Drug loading contet (%) = (Weight of the drug encapsulated in the DLP)/(Weight of the DLP) × 100%
Encapsulation efficiency (%) = ((Amount of the drug encapsulated in the DLP))/((Amount of the drug in feed)) × 100%

### 2.6. Characterization of DLP

The stability of DLP was detected by measuring the changes in particle size using DLS. Briefly, DLP was dissolved in pH 7.4 PBS, and the particle size was detected at various time points (0, 1, 3, 6, 12, 24, 48, 72, 96, and 120 h). For the drug leakage study, a DLP-containing PBS solution was placed in the dialysis bag (MWCO 6k-8kD) and immersed in 5 mL of 10-mM PBS (pH 7.4). At various time points (0, 1, 3, 6, 12, 24, and 48 h), 0.2 mL of external solution was taken, and fluorescence intensity was detected using an HPLC Fluorescence Detector at excitation wavelengths of 480 nm and 570 nm of emission wavelength for the released DOX. To study the drug release, DLP was suspended in PBS solution, transferred into a dialysis bag (MWCO 6k-8kD), and dialyzed against 5 mL of PBS buffers at pH 7.4 and pH 5. At various time points, the fluorescence intensity of the extracellular solution was measured using an multi-function microplate analyzer (TECAN Infinite 200, Männedorf, canton of Zürich, Switzerland) at 480 nm of excitation wavelength and 570 nm of emission wavelength for the released DOX. The accumulative release was calculated as: Cumulative release (%) = (DOX conc. in buffer solution)/(total DOX conc. in each sample) × 100%.

### 2.7. MTT Cytotoxicity of DLP

The in vitro cytotoxicity of DLP was investigated using HCT116 colon cancer cells and L929 fibroblast normal cells through an MTT assay, as per our previous protocol. Vitamin-B6, P(HPMA-*co*-MA-*hyd*-VB6), LP, DOX-loaded liposomes (abbreviated as DL), FAK siRNA, and free DOX were used for comparison. Briefly, HCT116 colon cancer cells and L929 fibroblast normal cells were seeded at a density of 2.5 × 10^4^ cells per well in 96-well plates and incubated for 24 h. The cells were then incubated with vitamin B6, P(HPMA-co-MA-hyd-VB6), DLP, DL, free DOX, LP, and FAK siRNA at 37 °C. After incubation for 48 h, all the groups after treatments (including the non-treatment group) were washed with PBS. After PBS washing, the MTT reagent-containing medium was immediately added into 96-well plates for reaction over 4 h. The medium in each well was removed, and DMSO was then added to dissolve the internalized purple formazan crystals. The absorbance was measured at the test wavelength (570 nm) and reference wavelength (633 nm) using a multi-function microplate analyzer (multimode microplate readers). The cells treated with the medium in the absence of drugs were defined as the control group with 100% viability. The cell viability (%) was determined by comparing the control group.

### 2.8. Endosomal Escape and Intracellular Drug Release

HCT116 colon cancer cells were seeded in a six-well plate at a density of 4 × 10^4^ cells/well and incubated at 37 °C under 5% CO_2_ for 24 h. The cells were then co-incubated with free DOX, LP, DLP, and free siRNA for 1, 3, and 6 h at 37 °C. The concentrations of DOX and siRNA were 10 μg/mL and 3.33 nM, respectively. After incubation, the cells were washed with PBS three times, stained with deep red Lysotracker for 1 h, and fixed with 4% formaldehyde over 30 min. The fixed cells were then washed with PBS again, followed by nuclear staining with DAPI Fluoromount-G™ mounting medium. Finally, cellular internalization was visualized using a confocal microscope (ZEISS LSM 800, Oberkochen, Germany). The excitation wavelengths for DOX and siRNA were 470 nm and 488 nm, respectively, while the emission wavelengths were 550 nm and 520 nm, respectively.

### 2.9. Western Blot Analysis

HCT116 colon cancer cells were cultured in a six-well plate at 37 °C and 5% CO_2_. After 24 h of incubation, DLP, DOX + siRNA, free DOX, and free siRNA were added. After 24 h, the cells were washed with PBS three times and lysed by adding RIPA lysis buffer. The lysed cells were centrifuged at 12,000 rpm for 30 min, and then a supernatant containing protein was extracted. The extracted protein was separated using polyacrylamide gel electrophoresis (SDS-PAGE) and transferred to the PVDF membrane. The PVDF membrane was blocked with 5% non-fat milk in TBS buffers overnight at 4 °C. The membrane was washed with a washing buffer three times and then incubated overnight at 4 °C with a monoclonal antibody against FAK and β-tubulin. Furthermore, the membrane was washed with washing buffer four times and incubated with a secondary antibody for 2 h. Finally, the membrane was observed by an enhanced chemiluminescence system as per the manufacturer’s instructions (Perkin Elmer, Waltham, MA, USA).

### 2.10. Biodistribution

First, 2.5 × 10^6^ cancer cells (HCT116) were subcutaneously seeded via inoculations in the front armpits of female BALB/c nude mice (6–8 weeks of age, weighing approximately 20 g). The experiment protocol was approved by the ethical committee for animal experiments at National Yang Ming Chiao Tung University. To observe the biodistribution of DLP, we first prepared DLP using synthesized cholesterol-NH2 instead of cholesterol and then conjugated Cy5.5 into DLP via a substitution reaction of amino groups from cholesterol with NHS ester from Cy5.5. The synthesis method of cholesterol-NH2 was the same as our previous report [36]. After 4 weeks of tumor growth (>500 mm^3^) and the tail vein of the tumor-bearing BALB/c nude mice was intravenously (i.v.) injected. The ex vivo fluorescent scans were performed by scarifying mice after 24 h of post-i.v. injection using the IVIS imaging system series 50 (Perkin Elmer, Waltham, MA, USA) with an excitation band filter at 682 nm and an emission at 702 nm.

### 2.11. In Vivo Anti-Tumor Efficacy Study

After 3 weeks of tumor growth (~200 mm^3^), the mice were treated with DLP, DL plus LP, or free DOX plus FAK siRNA via the lateral tail vein injection at a dose of 2 mg/kg for DOX and 10 μmol/kg for siRNA. A vernier caliper was used to measure the tumor sizes every 2 days, and volume was measured using the formula V = L × S^2^ × 0.5, wherein L and S represent the tumor dimension at the longest and smallest point, respectively. Relative tumor volumes were calculated as V/V_0_ (V_0_: volume of the tumor when the treatment was initiated). In addition, relative body weight was calculated as W/W_0_ × 100 (W_0_: body weight of mice when the treatment was initiated).

### 2.12. Statistical Analysis

Data are represented as the mean ± standard deviation. All results are representative of at least three sets of independent experiments, with samples performed in duplicate or triplicate in each experiment. The significance of the differences was determined using Student’s *t*-test, one-tailed, for each paired experiment. * *p*-value < 0.05 was considered statistically significant in all cases. * *p* < 0.05, ** *p* < 0.01, *** *p* < 0.001.

## 3. Results

### 3.1. Synthesis and Characterization of Polymer, Polymer-Vitamin B6 Conjugate, and mPEG-Cholesterol

This section may be divided into subheadings. It should provide a concise and precise description of the experimental results, their interpretation, and the experimental conclusions that can be drawn. In this study, several reaction steps were conducted to synthesize a biocompatible polymer, P(HPMA-*co*-MA-*hyd*-VB6), as shown in Table 1. ^1^H-NMR and FT-IR were used to confirm the structure of P(HPMA-*co*-MA-*hyd*-VB6). The ^1^H-NMR (DMSO-d_6_) characteristic peaks of P(HPMA-*co*-MA-*hyd*-VB6) showed a chemical shift for the hydrazone linker (-N-CH-) at δ 8.7–9.0; a shift for PHPMA (-N-CH_2_-C-) at δ 2.7–3.0; and a shift for pyridoxal (-O-CH_2_-C-) at δ 5.2–5.5 (Appendix A). Furthermore, FT-IR also showed a shift in the stretching wavenumber of aromatic compounds (-CH-) in pyridoxal at 3000 cm^−1^; a shift of hydrazone (-N = C-) at 1650–1700 cm^−1^; and a shift of carbonyl compounds in PHPMA at 1700–1750 cm^−1^ (Appendix A). Moreover, the GPC results also indicated the conjugation of VB6 on P10K45 copolymers (Appendix A). For the pH sensitivity of P10K45 copolymers, the titration results demonstrated that the copolymers not only absorbed protons and became hydrophilic but also had great water solubility at pH 3 and pH 10 (Appendix A). These analysis results indicated that P(HPMA-*co*-MA-*hyd*-VB6) copolymers were successfully synthesized as designed. 

On the other hand, 1H-NMR and FT-IR were also used to identify the structure of mPEG-Chol. ^1^H-NMR (CDCl_3_) showed a chemical shift of mPEG(-O-CH_2_-CH_2_-) at δ 3.4–4.2 and a shift of cholesterol (-CH = C-) at δ 5.3–5.5 (Appendix A). FT-IR indicated a wavenumber of cholesterol (-C = C-) at 2900 cm^−1^ and a wavenumber of mPEG (-C-O-) at 1300 cm^−1^ (Appendix A). These analysis results also revealed the successful formation of mPEG-Chol. The degree of purity for mPEG-Chol was calculated as 99% by considering the characteristic peak of cholesterol at δ 5.3–5.5 ppm and the mPEG at δ 3.2–3.3 ppm.

### 3.2. Formation and Characterization of LP and DLP

This section may be divided into subheadings. It should provide a concise and precise description of the experimental results, their interpretation, and the experimental conclusions that can be drawn. Before the preparation of LP and DLP, searching a forming condition for polyplexes from P(HPMA-*co*-MA-*hyd*-VB6) copolymers and siRNA was investigated at various N/P ratios after incubating them at pH 4.0 and pH 7.4 for 20 min and 24 h. The P40K45 was used as a testing sample in this study. As revealed in Figure 2A, P40K45 failed to form polyplexes at pH 7.0. Conversely, as shown in Figure 2B, P40K45 forms stable polyplexes within 20 min as the N/P ratio increased to 200 when the pH was at 4.0. Furthermore, to investigate the effects of molecular weight and degree of pyridoxal of copolymers on the formation of polyplexes, P(HPMA-*co*-MA-*hyd*-VB6) copolymers with various compositions were mixed with siRNA. As shown in Figure 2C, as the molecular weight of copolymers increased, the adsorption of siRNA on copolymers was weak. In contrast, as the pyridoxal ratio in the copolymer increased, the tendency to form polyplexes with around 100% of the siRNA encapsulation efficacy after incubation for 20 min was observed at a N/P ratio of 200. Although instability and dissociation of polyplexes were observed as the formation time of polyplexes increased to 24 h (Figure 2D), polyplexes were then immediately encapsulated into liposomes without causing siRNA dissociation. As P10K45 had higher siRNA affinity than other copolymers, it was selected for further lipopolyplex preparation and function evaluation.

Lipopolyplexes without DOX content were prepared via a three-step process, i.e., thin-film hydration, sonication, and filtration, from siRNA-P10K45 interacted polyplexes, DPPC phospholipids, and mPEG-chol copolymers. DOX was loaded into LP using the sodium ammonium sulfate gradient method to form DOX-loaded lipopolyplexes (DLP). The particle size of synthesized LP and DLP was investigated using DLS. As summarized in Table 2, the particle sizes of synthesized LP and DLP were approximately 191 nm (PDI of 0.1) and 168 nm (PDI of 0.1), respectively (Appendix A). For the reason that the DOX loaded into DLP using a sodium ammonium sulfate gradient method could become DOX-sulfate crystals, we predicted that the inner core of DLP was denser than that of LP, resulting in a decrease in particle size when encapsulating DOX. The drug loading (D.L.) contents of DOX for LP and DLP were 0 and 4 wt%, respectively. In addition, the encapsulation efficiency (E.E.) of DOX for LP and DLP were 0 and 46%, respectively. Moreover, the DLP were stable at 25 °C and 37 °C for 120 h (Figure 3A). Furthermore, the morphologies of LP and DLP were observed using TEM, as shown in Figure 3B, respectively. The particle sizes of the LP and DLP obtained by TEM also agreed with the DLS results.

The in vitro drug release study was conducted at 37 °C in PBS at pH 7.4 and 5.0, and the amount of DOX released from DLP was measured at the predetermined time intervals using ELISA readers at an excitation wavelength of 480 nm and an emission wavelength of 570 nm. As shown in Figure 3C, the maximum DOX release was observed at a lower pH than the physiological pH value. Approximately 55% and 19% of DOX were released after 168 h of incubation at pH 5.0 and pH 7.4, respectively. Furthermore, TEM results show that there is no change in the particle size distribution of DLP at physiological pH (Figure 3D), whereas the morphology of DLP changed and the liposomal bilayer was broken at pH 5.0 (Figure 3D).

HCT116 colon cancer cells were co-cultured with FAM dye-labeled siRNA, DLP within siRNA, and LP within FAM dye-labeled siRNA for 1, 3, and 6 h. LysoTracker dye was also used to stain acidic components, including endosomes and secondary lysosomes. As shown in Figure 4A, free FAM dye-labeled siRNA was difficult to internalize into cancer cells. In contrast, the fluorescent intensities of released siRNA and DOX were observed in cancer cells and separated from those for LysoTracker at 1, 3, and 6 h (Figure 4 and Figure 5).

### 3.3. Biocompatibility and Cytotoxicity Study

This section may be divided into subheadings. It should provide a concise and precise description of the experimental results, their interpretation, and the experimental conclusions that can be drawn. As shown in Figure 6A,B, copolymers showed less cytotoxicity than free VB6 molecules for L929 fibroblast normal cells as well as HCT116 colon cancer cells. However, copolymers could reduce the toxicity of VB6 molecules because VB6 is conjugated with copolymers. Furthermore, the cytotoxicity of free DOX, free siRNA, LP, DOX-loaded liposomes (DL), and DLP were assessed against L929 normal cells and HCT116 colon cancer cells (Figure 6C–F). DLP was more toxic than either free DOX or DL for cancer cells, whereas it showed relative safety as compared with free DOX for normal cells. Furthermore, LP showed higher cytotoxicity than free siRNA at the same siRNA concentration for normal cells as well as cancer cells.

Western blotting was carried out in order to investigate the inhibitory effects of DLP on FAK protein synthesis. The siRNA used in this study is an mRNA interference fragment that inhibits the translation of FAK proteins. As shown in Figure 7A, the expression of FAK protein was enhanced in the presence of free DOX and DOX + siRNA groups in comparison with the control and other treatments. Most interestingly, the FAK protein content was reduced in the presence of DLP in comparison to the control, free siRNA, and DOX + siRNA groups, which proved that the DLP successfully delivered siRNA into the cytoplasm to inhibit the synthesis of FAK proteins.

The cell cycle analysis was investigated using HCT116 cancer cells after treatment with free DOX, copolymers, LP, and DLP (at DOX and siRNA concentrations of 20 μg/mL and 10 nM, respectively). As revealed in Figure 7B, the copolymers and LP, as well as the control group, arrested HCT116 cancer cells at the G0G1 phase. However, HCT116 cancer cells treated with free DOX and DLP showed a higher number of cells at the S phase, whereas the number of cells at G0G1 phases decreased compared with the untreated control group, indicating that the cell cycle arrestment was controlled by DOX.

### 3.4. Ex-Vivo Biodistribution and In Vivo Tumor Growth Inhibition Study

An ex-vivo biodistribution study was investigated after 24 h of post i.v. injection of Cy5.5 labeled DLP into HCT116 tumor-bearing BALB/c nude mice using an IVIS imaging system. As shown in Figure 8A, high fluorescence intensity was observed in the tumor tissue in comparison to other organs. Based on the relative intensity, around 25% of DLP was accumulated in the tumor regions. Furthermore, the therapeutic efficacy of DLP, free drugs (DOX + siRNA), and DL + LP were investigated using BALB/c nude mice containing HCT116 tumor cells. As revealed in Figure 8B,C, the control group showed fast tumor growth as compared with the other groups. Most importantly, DLP showed significantly better tumor growth inhibition than free drugs (DOX + siRNA) and DL + LPI, owing to the co-delivery of DOX and siRNA using a single nanocarrier that could enhance therapeutic efficacy either through synergistic or additive effects. As shown in Figure 8E, there is no significant change in the percentage of body weight. Furthermore, the images of tumors isolated on day 30 verified that mice treated with DLP had the smallest tumor size (Figure 8D).

A biochemical index was also tested in order to ensure the biocompatibility of synthesized materials after intravenous injection. The two most common kidney function tests, blood urea nitrogen (BUN) and creatinine, were conducted using BUN and creatinine kits. As shown in Figure 8E, BUN and creatinine values lay within the normal ranges for all groups (BUN 10~33 mg/dL and creatinine 0.5~2.2 mg/dL), indicating that synthesized materials could not affect kidney function. Similarly, liver function tests were conducted by measuring GPT and GOT values. As shown in Figure 8F, the levels of GPT and GOT in the treating groups were also within the normal range (GPT 28–132 U/L and GOT 59–247 U/L) and lower than those in the control group, indicating that the synthesized material has no effects on the liver.

## 4. Discussion

In recent years, drug delivery systems have been keenly explored to enhance the therapeutic efficiency of pharmaceutical agents and minimize their side effects on normal cells. A critical concern in drug delivery systems is the synthesis of biocompatible nanocarriers. Lipopolyplexes, which are well-known nanocarriers having a core-shell structure composed of lipids, polycations, and nucleic acids, show superior colloidal stability, low cytotoxicity, and high gene transfection efficiency as they combine the advantages of polyplexes and lipoplexes. To achieve excellent cancer chemo/gene combination therapy, we developed a pH-sensitive charge-conversion copolymer, P(HPMA-*co*-MA-*hyd*-VB6), which bears positive charges at pH 4 and neutral charges at pH 7.4, to facilitate electrostatic interaction with negatively charged FAK siRNA for preparing polyplexes. The polyplexes and the anticancer drug DOX can then be encapsulated into PEGlyated liposomes to promote FAK siRNA release efficiency, prevent side effects caused by DOX and cationic polymers, and improve efficient endosome escape of FAK siRNA and DOX.

As is well-known, the N/P ratio is an important factor for gene delivery. A high N/P ratio of polyplexes is often highly cytotoxic to normal tissues because a large number of positive charges to polymer carriers could disrupt cell membranes [37], induce cell necrosis [38], and damage mitochondria [39]. In our study, the FAK siRNA absorption ability for P(HPMA-*co*-MA-*hyd*-VB6) copolymers arose from the protonation of pyridine of VB6 molecules. Most interestingly, in our in vitro study, as the concentration of free VB6 increased, so did cytotoxicity. Similar results were also reported for VB6 by other researchers using diverse types of cancer cell lines [40,41,42]. As VB6 was conjugated to the copolymers, the toxicity induced by VB6 was reduced. In addition, P(HPMA-*co*-MA-*hyd*-VB6) copolymers had pH-sensitive charge-conversion ability; they bore neutral charges at pH 7.0 and positive charges below pH 4.2. Although the N/P ratio of P(HPMA-*co*-MA-*hyd*-VB6) copolymers was nearly 200, which was much higher than the reasonable ratio for gene therapy, P(HPMA-*co*-MA-*hyd*-VB6) copolymers and P(HPMA-*co*-MA-*hyd*-VB6)-containing lipopolyplexes exhibited low toxicity in our in vitro and in vivo tests because copolymers could be converted to non-toxic neutral-charged form after they were delivered into cell cytosols and blood.

Along with cytotoxicity, gene loading and release abilities are also key factors of nanocarriers for gene delivery. To obtain better gene loading efficiency, cationic polymers with hydrophobic segments have been developed to densely condense negatively charged genes, enhancing gene loading efficiency [43,44]. In addition, hydrophobic segments have been found to promote the transfection efficiency of cationic polymers [45]. Furthermore, gene escape from endosomes is a critical step for successful transfection. Several reagents, such as photosensitizers [46], polymers [47], and peptides [48], have been used to promote endosomal escape. Among these, although endosome-disrupting peptides have emerged as a promising approach because they could overcome lysosomal entrapment and degradation of genes [49], our P(HPMA-*co*-MA-*hyd*-VB6) copolymers could become hydrophobic at pH 4.0 for stabilizing FAK siRNA/copolymer complexes during lipopolyplex preparation, transform to being hydrophilic at pH 7.4 for dissociating FAK siRNA in the core of lipopolyplexes and cell cytosols, and induce lipopolyplex disruption and endosomal escape owing to the large amount of VB6 in copolymers. Therefore, lipopolyplexes showed rapidly endosomal escape ability after being cell-internalised for 1 h and inhibited the synthesis of FAK proteins.

FAK is a protein tyrosine kinase that is overexpressed in various tumor types and associated with the poor prognosis of cancer patients, such as gastric cancer (Appendix A) [50]. Recently, the discovery of FAK inhibitors has gained great interest. Although several strategies for inhibiting FAK protein expression, such as interfering with the ATP-binding pocket of FAK and inhibiting the phosphorylation of FAK, have been developed [51], combinational therapy with other therapeutics is still the dominant method because FAK inhibitors are insufficient to fully treat cancer [52,53]. In this study, we co-encapsulated FAK siRNA and DOX into lipopolyplexes to maximize the functionality of a FAK inhibitor and an anticancer drug in cancer therapy. Moreover, liposome-based nanomedicines have been reported as a stable and suitable carrier to prevent RNase degradation of siRNA in the blood circulation and to increase the tumor accumulation of siRNA in the tumor-bearing mouse model [54]. In our in vitro test, free DOX treatment could up-regulate the FAK expression in HCT116 colon cancer cells. Nevertheless, DOX combined with free FAK siRNA slightly reduced FAK expression. The VB6-based lipopolyplexe co-delivery system not only effectively transported DOX and transfected FAK siRNA into the cytosols of cancer cells but also significantly downregulated the expression of FAK protein. For the reason that P(HPMA-*co*-MA-*hyd*-VB6) copolymers induced lipid bilayer disruption at low pH, the loaded FAK siRNA and DOX could simultaneously and precisely release in the cytoplasm for acting on a single cancer cell. Therefore, lipopolyplexes showed better in vivo therapeutic efficacy than DOX-loaded liposomes + FAK siRNA-loaded lipopolyplexes. Although FAK siRNA-loaded lipopolyplexes still induced toxicity in normal cells, the cytotoxicity of free DOX was much higher than that of DOX-loaded liposomes and lipopolyplexes. Hence, normal cell cytotoxicity could be overcome by co-encapsulating DOX and FAK siRNA into lipopolyplexes as compared with free DOX.

## 5. Conclusions

In this study, polyplexes were successfully synthesized using P(HPMA-*co*-MA-*hyd*-VB6) copolymers and FAK siRNA in pH 4.0 surroundings via electrostatic interactions. Polyplexes and DOX were co-encapsulated into mPEG-cholesterol modified liposomes to form lipopolyplexes. DOX and siRNA release studies revealed that P(HPMA-*co*-MA-*hyd*-VB6) copolymers could respond to low pH and induce the disruption of lipopolyplexes and endosomes for releasing DOX and siRNA. Therefore, lipopolyplexes could cause high cytotoxicity and reduce the translation of FAK proteins into cancer cells. In addition, lipopolyplexes could also accumulate in the tumors and inhibit tumor growth without affecting liver and kidney functions. In general, this synthesized nanocarrier, lipopolyplex, is believed to be a suitable candidate for the co-delivery of chemotherapeutic agents and genes to enhance the therapeutic efficacy of cancer treatments.

## Figures and Tables

**Figure 1 pharmaceutics-15-00596-f001:**
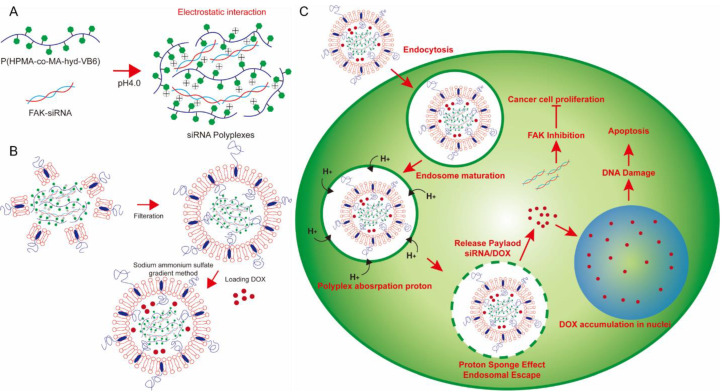
The concept of DOX-loaded lipopolyplexes for cancer chemo-gene therapy. (**A**) At pH 4.0, P(HPMA-*co*-MA-*hyd*-VB6) copolymers could absorb protons to become positive charges for electrostatic interaction with negatively charged FAK siRNA to form gene-loaded polyplexes. (**B**) The gene-loaded polyplexes were then encapsulated and DOX were loaded into PEGlyated-liposomes using a sodium ammonium sulfate gradient method to form lipopolyplexes. (**C**) After cancer cells internalized lipopolyplexes via an endocytosis process, P(HPMA-*co*-MA-*hyd*-VB6) copolymers induced proton sponge effect and simultaneously deformed the structures of lipopolyplexes and endosomes for releasing FAK siRNA and DOX into cell cytosols for cancer chemo-gene therapy.

**Figure 2 pharmaceutics-15-00596-f002:**
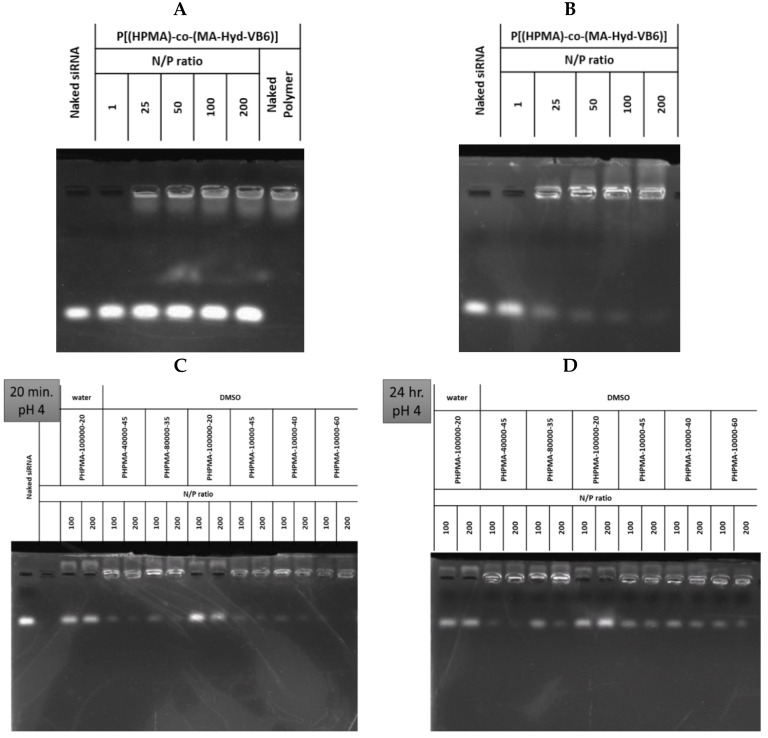
The absorption ability of P(HPMA-co-MA-hyd-VB6) copolymers with FAK siRNA. The northern blotting of P40K45 copolymers mixed with FAK siRNA under various nitrogen/phosphorus (N/P) ratios at (**A**) pH 7.0 for 24 h and (**B**) pH 4.0 for 20 min. The northern blotting of copolymers with various molecular weight and VB6 contents mixed with FAK siRNA at pH 4.0 for (**C**) 20 min and (**D**) 24 h. Water: the copolymers dissolved in the DMSO/pH 7.4 PBS solution (*v*/*v* = 1:1). DMSO: the copolymers dissolved in the DMSO/pH 4.0 300 mM citrate buffer (*v*/*v* = 1:1).

**Figure 3 pharmaceutics-15-00596-f003:**
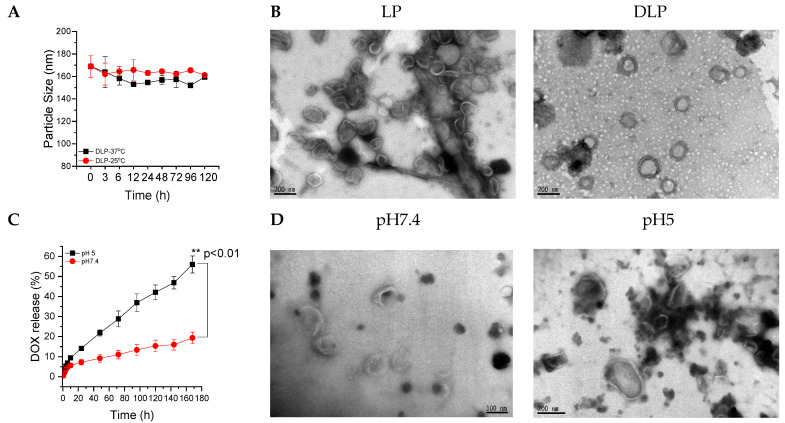
(**A**) The particle size of DLP at 25 °C and 37 °C surroundings. (**B**) TEM images of LP and DLP lipopolyplexes. The scale bars were 200 nm. (**C**) Accumulative release of DOX from DLP at pH 7.4 and 5.0. (**D**) TEM image of DLP after 6 h of incubation at pH 7.4 and pH 5.0. The scale bars for left and right panel were 100 and 200 nm, respectively. Data were expressed as mean ± SD values (*n* = 3). ** *p* < 0.01.

**Figure 4 pharmaceutics-15-00596-f004:**
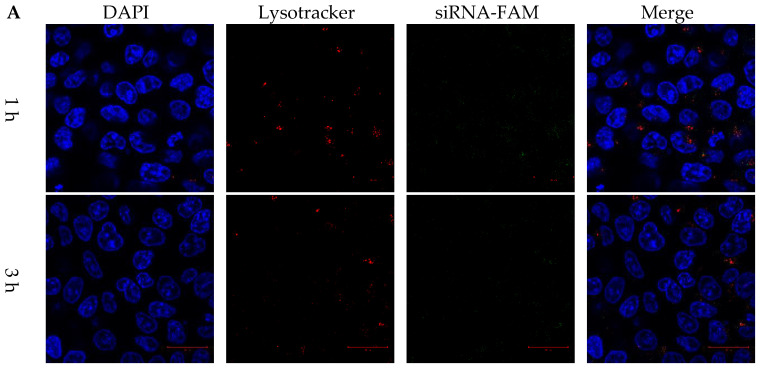
Cytosol delivery behaviors of siRNA from lipopolyplexes. Confocal images of HCT116 cancer cells treated with (**A**) free siRNA and (**B**) LP for 1, 3 and 6 h. The blue, red and green colors represented DAPI-stained nuclei, lysotracker stained lysosome, and siRNA-FAM, respectively. The white arrow indicated that the release of siRNA. The scale bars were 20 μm.

**Figure 5 pharmaceutics-15-00596-f005:**
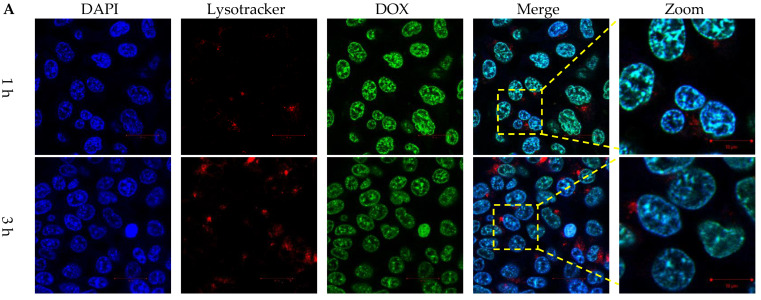
Intracellular DOX release behaviors of lipopolyplexes. Confocal images of HCT116 cancer cells treated with (**A**) free DOX and (**B**) DLP for 1, 3 and 6 h. The blue, red and green colors represented DAPI-stained nuclei, lysotracker stained lysosome, and DOX, respectively. The scale bars were 20 µm.

**Figure 6 pharmaceutics-15-00596-f006:**
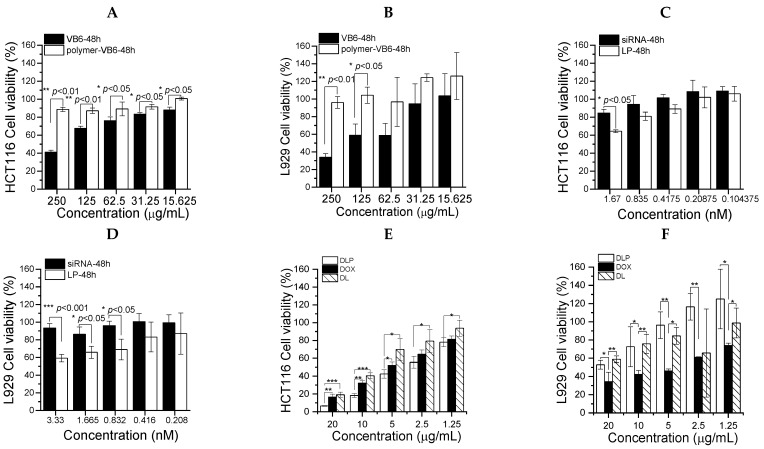
Cytotoxicity of lipopolyplexes in HCT116 cancer cells and L929 fibroblasts. (**A**) HCT116 cells and (**B**) L929 fibroblasts treated with VB6 and P(HPMA-co-MA-hyd-VB6) (Polymer-VB6) for 48 h. Free FAK siRNA and LP incubated with (**C**) HCT116 cells and (**D**) L929 for 48 h. (**E**) HCT116 cells and (**F**) L929 fibroblasts treated with free DOX, DL and DLP for 48 h. Data were presented as mean ± SD (*n* = 3). * *p* < 0.05, ** *p* < 0.01 and *** *p* < 0.001.

**Figure 7 pharmaceutics-15-00596-f007:**
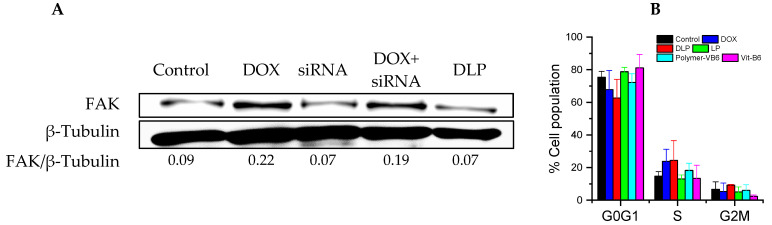
Bio-activity of lipopolyplexes. (**A**) Western blotting of HCT116 cancer cells after treating with each sample for 24 h. FAK/β-Tubulin ratio was calculated using imageJ software. (**B**) Cell cycle analysis of PI-stained HCT116 cancer cells. Cells were incubated with each sample for 24 h. DOX: free DOX (20 μg/mL); siRNA: FAK siRNA (10 nM); DOX + siRNA: DOX with FAK siRNA (Dox: 20 μg/mL; FAK siRNA: 10 nM). Data were presented as mean ± SD (*n* = 3).

**Figure 8 pharmaceutics-15-00596-f008:**
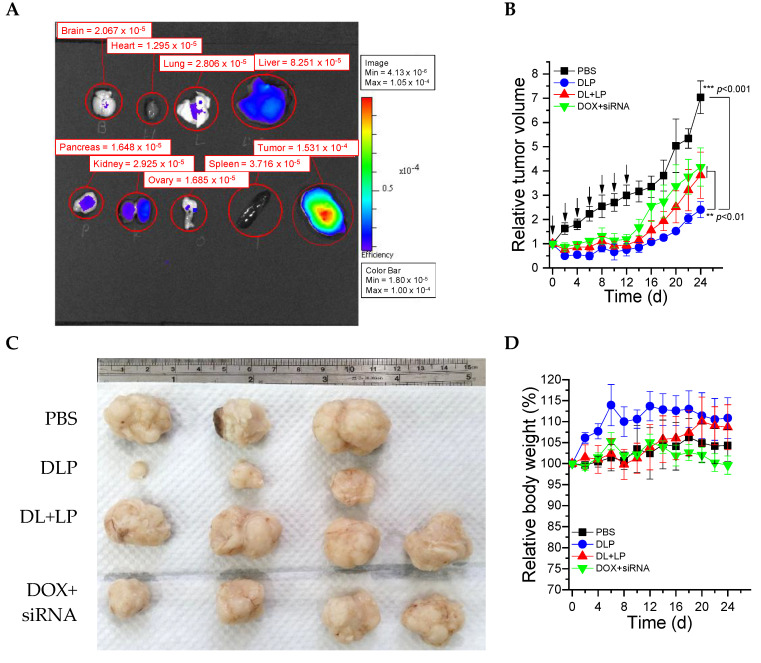
In vivo test of lipopolyplexes. (**A**) The biodistribution of Cy5.5-labeled lipopolyplexes in each organ after treatment for 24 h. (**B**) Relative tumor volume changes of BALB/c nude mice after treatment for 24 days. (**C**) Images of real tumor volume after treatment for 24 days. (**D**) Relative body weight changes of BALB/c nude mice after treatment for 24 days. (**E**) Kidney function and (**F**) liver function of BALB/c nude mice after treatment for 24 days. The drugs were administered on day 0, 2, 4, 6, 8, 10, and 12. PBS: mice treated with PBS; DLP: mice treated with DLP; DL+LP: mice treated with DOX-loaded liposomes and lipopolyplexes; DOX + siRNA: mice treated with free DOX and free FAK siRNA. The treated dosages of DOX and siRNA were 2 mg/kg and 10 µmol/kg, respectively. Data were presented as mean ± SD (*n* = 4). ** *p* < 0.01 and *** *p* < 0.001.

**Table 1 pharmaceutics-15-00596-t001:** The composition of P(HPMA-*co*-MA-*hyd*-VB6) copolymers.

Code	In Feed (mol %)			In Copolymer (Mol %) ^a^		Copolymer Property ^b^		
	HPMA	MABH	MPA	HPMA	Pyridoxal	Mw	Mn	PDI
P40K45	46.5	46.5	7	53.7	46.3	43,700	15,700	2.78
P80K35	66.7	33.3	0	65.0	35.0	78,300	13,300	5.88
P100K20	84.2	15.8	0	83.2	16.8	115,500	23,100	5.00
P10K45	45.5	45.5	9	55.9	44.1	11,400	2600	3.98
P10K60	22.7	62.1	9.2	39.3	60.7	9300	2600	3.60

^a^ The composition of copolymers was calculated from ^1^H-NMR. ^b^ The weight-average molecular weight (Mw), number-average molecular weight (Mn) and polydispersity index (PDI) were determined using GPC.

**Table 2 pharmaceutics-15-00596-t002:** The characterization of LP and DLP at 25 °C.

Code	Size (nm) ^a^	PDI ^a^	E.E. (%) ^b^	D.L. (wt %) ^b^
LP	190.9 ± 27.25	0.099 ± 0.048	0	0
DLP	167.77 ± 7.08	0.095 ± 0.035	45.69 ± 9.49	4.01 ± 0.61

^a^ The particle size and size distribution (PDI) of LP and DLP were determined by DLS. ^b^ The drug content and encapsulation efficiency of Dox for LP and DLP were measured using UV-Vis spectrophotometer at 480 nm wavelength.

## Data Availability

Data is contained within the article.

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
