# Peer review of "Lipopolyplex-Mediated Co-Delivery of Doxorubicin and FAK siRNA to Enhance Therapeutic Efficiency of Treating Colorectal Cancer"

_pharmaceutics, 2023, doi:10.3390/pharmaceutics15020596_

Round 1
Reviewer 1 Report
In this manuscript, the authors report the development of DOX and FAK siRNA co-delivery systems to enhance therapeutic efficiency for colon cancer treatment. There are some issues that the authors need to address before it can be reconsidered for publication.
1. Novelty of the study should be clearly mentioned in the introduction. I think that there are many similar researches about Dox and siRNA co-delivery systems. Compared with previous research, the authors should describe the advantages of their approach based on not only concept but also the obtained results in the introduction and discussion. It is not clear that they represent a major advance over what is already in the literature.
2. In Figure 3A, please revise the unit of temperature “oC”.
3. In Figure 3D, the scale bar was presented as 100 nm, but the authors describe “The scale bars were 200 nm” in the figure caption. Please revise it.
4. In Figure 6, it is suggested that the authors should change the scale of the y-axis of Figure 6A to match Figure 6B for better comparison. For the same reason, please change the scale of the y-axis of Figure 6C, 6D, 6E and 6F.
5. There are many problems about superscript and subscript of units (such as cm-1, oC, DMSO-d6, CDCl3 and so on). I believe these are only part of them, please spend a little more time to improve the manuscript.
Reviewer 2 Report
1. Title should not include “cells”.
2. What about the siRNA release from the DLP.
3. How to make sure all the byproducts were removed completely.
4. Did the author investigate how much of siRNA will be lose when they prepared the LPs? What about the stability during the process.
5. Is there any way that can measure the pH of the inner part of that formulation, how to keep the inner is 4, and outside is 7.4? Please write down how to prepare the LP.
6. When the author investigated the MTT assay, why washing with PBS before adding MTT? PBS washing might remove some of the dead cells and affect the results and conclusion. For the cell viability study, the cells that treated by PBS (untreated group) should be used as the control (100%).
7. The excitation and emission wavelength of DOX and FAM-siRNA are very close. It’s better to use Cy3-siRNA or Cy5.5-siRNA inhere to investigate the cellular uptake.
8. Western blot study should have the DL+LP group.
9. How to conjugate Cy5.5 into DLP? Please include this in the methods part. How’s the siRNA biodistribution in tumor mice?
10. Please confirm the in vivo siRNA dose, 10 nM/kg? It’s impossible.
11. Line 291. Why the polymer become hydrophobic under 4?
12. The authors investigated the siRNA encapsulation, what about the siRNA transfection?
13. Figure 2, please explain why the initial wells showed the bright band.
14. Why the particle size of DLP exhibited lower than LP? Usually, the particle size will be increase when encapsulate drug.
15. TEM images are not clear, please repeat that.
16. Cellular uptake images are not clear. Flow cytometry should be used to investigate the cellular uptake. Flow is much more acceptable than confocal. Figure 5 showed that the DLP decreased the DOX cellular uptake when compared with free DOX, so, what the means of encapsulate the DOX into DLP in here.
17. Why the FAK protein was enhanced in the presence of free DOX and DOX +siRNA?
18. As shown in Figure 7A, the free siRNA group also showed the inhibition of FAK expression, compared with the DLP, is there any statistical difference?
19. Figure 8A is worthless in here. It’s better to compare the biodistribution of DOX in DLP, DL+LP, and DOX, then investigate if there is any improvement or not.
Reviewer 3 Report
This is an interesting paper. The authors should comment on the following:
1. Why they separated the free dox by a sephadex column rather than a dialysis procedure?
2. In Suplementary Material, the NMR of figure S3 is not consistent, after the conjugation of VB the protons at 5.5 ppm should be 6 in respect to f protons not 1.
3. No regression of tumors was achieved.
4. The liver of the animals was influenced by the nanoparticles.
5. DLS measurements should be shown in Supplementary material.
6. On Figure S4, at 2880 cm-1 should be aliphatic CH rather than aromatic bonds.
I think it should be published in another journal.
Reviewer 4 Report
In this paper, the authors prepared a zwitterionic polymer to form a complex with FAK siRNA. The complex and DOX were further encapsulated into liposomes for the treatment of colon cancer. The use of Lipopolyplex for co-delivery of RNA and chemical drug is interesting. However, there are some concerns need to be addressed before it can be considered for publication.
1. In method section, preparation of lipopolyplexes (LP) and Dox-loaded lipopolyplexes (DLP), after the thin film was hydrated using 1.5 mL of polyplex-containing citrate buffer, diethyl ether was added. Did diethyl ether break the liposome structure that formed in the hydration?
2. In figure 2C and 2D, what is the different between water and DMSO group? The authors should provide more specific figure captions.
3. In figure 3, what is the particle size, Zeta potential, PDI and morphology of the zwitterionic polymer-siRNA complex? There is no core shell structure observed in DLP group.
4. In cell trafficking study, the study should incubate the cells with the DLP for a fixed time and image the DLP trafficking in cell for a period. In this way, there won't be new taken DLP interfere with the results. The authors should also provide zoomed images. Besides, the free DOX looks interacting with the nucleus but the DOX from DLP located in cytoplasm.
5. In in vivo distribution study, why the in vivo image in figure 8A show no significant difference between tumor and other tissue? In Ex-vivo fluorescence images in figure 8B, more mice should be used to plot the statistical bar chart. It was reported that the average tumor delivery efficiency is 0.7%, 25% is a high tumor accumulation. What is the reason that cause the high tumor targeting efficiency of the DLP.
Round 2
Reviewer 2 Report
Comment 5: The authors didn't answer that question properly. please answer how to keep the inner pH of the formulations is 4.
Comment 6: Please repeat the MTT assay without washing with PBS, just replace the medium with PBS prior to starting the MTT measurement. In figure 6, the cell viability study, which group is the 100% cell viability group? It's better to make sure have the control (PBS treated group) as the 100% viability group.
Comment 9: Free siRNA will be rapidly degraded by RNase in the blood, what about your formulations? the authors can evaluate the siRNA distribution compare with the free siRNA and formulation to figure out if there is any improvement.
Comment 10: the dose should be 10 umol/kg, not 10 uM/kg. uM is a concentration unit, which is the umol per liter. please correct that.
Comment 12: MTT cannot indicate the siRNA encapsulation efficacy.
Comment 16: Please use the flow to evaluate the cellular uptake.
Comment 19: Please remove Figure 8A.
Reviewer 3 Report
It can be published as it is.
Reviewer 4 Report
The authors solved most of the concerns.
Round 3
Reviewer 2 Report
Please use Flow cytometry to evaluate the siRNA/DOX cellular uptake compared with free siRNA/free DOX, LP, and DLP. Figure 4 and 5 are as clear as mud. Flow is much more accuracy than just confocal images. Although marked by the white arrow, it's not convinced that LP improved siRNA uptake compared with free siRNA from Fogure 4. Authors should involve the Flow data no matter what the data is.
